# Gold Nanoparticles-Mediated Photothermal Therapy of Pancreas Using GATE: A New Simulation Platform

**DOI:** 10.3390/cancers14225686

**Published:** 2022-11-19

**Authors:** Somayeh Asadi, Leonardo Bianchi, Martina De Landro, Paola Saccomandi

**Affiliations:** Department of Mechanical Engineering, Politecnico di Milano, 20156 Milan, Italy

**Keywords:** GATE, thermal therapy, gold nanorods, pancreas cancer, Monte Carlo, simulation

## Abstract

**Simple Summary:**

Laser irradiation of noble metal nanoparticles has recently been investigated to achieve a potential clinically relevant photothermal effect. This work presents the first study of gold nanorods (GNRs)-assisted therapy of a pancreas tumor using the Monte Carlo-based code developed with Geant4 Application for Emission Tomography (GATE). Having the possibility of setting the optical and thermal properties of nanoparticles, GATE platform allows for the modeling of the GNRs-mediated hyperthermia therapy as well as heat diffusion in tissue. The code was benchmarked to validate the heat distribution between the simulation and the experiment.

**Abstract:**

This work presents the first investigation of gold nanorods (GNRs)-based photothermal therapy of the pancreas tumor using the Monte Carlo-based code implemented with Geant4 Application for Emission Tomography (GATE). The model of a human pancreas was obtained by segmenting an abdominal computed tomography (CT) scan, and its physical and chemical properties, were obtained from experimental and theoretical data. In GATE, GNRs-mediated hyperthermal therapy, simple heat diffusion as well as interstitial laser ablation were then modeled in the pancreas tumor by defining the optical parameters of this tissue when it is loaded with GNRs. Two different experimental setups on ex vivo pancreas tissue and GNRs-embedded water were devised to benchmark the developed Monte Carlo-based model for the hyperthermia in the pancreas alone and with GNRs, respectively. The influence of GNRs on heat distribution and temperature increase within the pancreas tumor was compared for two different power values (1.2 W and 2.1 W) when the tumor was exposed to 808 nm laser irradiation and with two different laser applicator diameters. Benchmark tests demonstrated the possibility of the accurate simulating of NPs-assisted thermal therapy and reproducing the experimental data with GATE software. Then, the output of the simulated GNR-mediated hyperthermia emphasized the importance of the precise evaluation of all of the parameters for optimizing the preplanning of cancer thermal therapy. Simulation results on temperature distribution in the pancreas tumor showed that the temperature enhancement caused by raising the power was increased with time in both the tumor with and without GNRs, but it was higher for the GNR-load tumor compared to tumor alone.

## 1. Introduction

Thermal treatments are a class of medical modality for cancer treatment, based on the increase in tumor temperature until necrosis and/or apoptosis occur. Thermal treatments have received much attention as minimally invasive strategies for the removal of focal malignant diseases by means of thermal energy sources, such as radiofrequency, microwaves, near-infrared (NIR) laser radiation, and high-intensity focused sonography [1]. Depending on the kind of thermal energy that is applied and the thermal response of the target tissue, direct cellular damage occurs at several levels, from the subcellular level to the tissue one. Based on the treatment duration and the tissue temperature, the outcome of thermal therapy can be categorized as it was by Habash et al. [2]. Heating the cells up to 41 °C is generally used in physiotherapy for the treatment of rheumatic diseases (i.e., diathermia). Raising the temperature to higher-than-normal body temperature (in the hyperthermia range, 41–45 °C) can be used to increase the effectiveness of other types of cancer treatments (such as radiation therapy or chemotherapy), thus enabling a combined treatment approach. When the temperature reaches above 45 °C and it is maintained for an adequate amount of time, the cells can be locally destroyed within a tumor. It can be used in oncology for cancer treatment.

Among the different energy-based modalities for the treatment of focal malignancies, light-based therapies, such as NIR laser treatments, have shown promising results. However, concerning this, the use of local temperature increases to treat cancer of the internal organs has encountered some difficulties in the optimal control of the heat delivery in the target, mostly due to the lack of selectivity of tumors with respect to the laser light. This lack of sensitivity can lead to two main problems: the under-treatment of the tumor, thus causing tumor recurrences, or the over-treatment, which may provoke unwanted thermal damage to the surrounding healthy structures. To address these challenges, combinational nanoparticles (NPs)-hyperthermia cancer therapy has been proposed as a solution to obtain a localized temperature increase with consequent irreversible tissue damage limited to the tumor with a low thermal dose. In this regard, NPs-based localized enhancement photothermal effect in tumors has been widely investigated [3,4,5], and, in particular, physically enhanced therapeutic effect by high-Z metal NPs-mediated radiosensitization has been reported in multiple pre-clinical studies [6,7,8,9,10].

Among the different NPs that have been designed for this purpose, rod-shaped gold NPs, i.e., gold nanorods (GNRs), are known to be effective agents for photothermal therapy (PTT), as they enhance therapeutic ratio and thus decrease the treatment time [11,12]. GNRs can strongly absorb electromagnetic waves at different frequencies corresponding to NIR radiation and can also act to induce a rapid temperature increase localized in tumors [11,13].

Several works have investigated the photothermal effects of GNRs in laser therapy of cancers and reviewed the most recent scientific progress, including the efficacy, molecular mechanism, and toxicity of GNPs-mediated photothermal therapy of cancers, which some of them limited to the in vitro and in vivo studies [14,15,16,17]. For instance, Mooney and colleagues performed an in vivo study on flank xenograft tumors to investigate the effect of factors such as the power of the laser source and NPs concentration, on the temperature reached on the surface and inside of the tumor [16]. In this work, a significant increase in the difference between the temperature of the surface and the internal part of the tumor was reported as a result of a change in NPs concentration. Popp et al. studied the efficacy of GNRs for PTT in a difficult and fast-growing murine melanoma model using a NIR light-emitting diode light source. The results showed that laser irradiation can effectively heat in vitro and in vivo models to ablative temperatures when combined with GNRs [17].

By considering the size effect on the amount of heat generated by GNRs in PTT, a diameter of 10 nm and the length of 40 nm, in which the longitudinal plasmon resonance occurs at a wavelength of around 808 nm, represents one of the most common sizes for these particles [18,19].

Despite many signs of progress in NPs-assisted PTT of cancer, this multifunctional approach remains a method with limited clinical applications since some obstacles hinder the translation from research to clinical practice. Furthermore, there is still a need to evaluate many factors such as the optimum concentration and size for NPs (by considering the kind of NPs) and thermal behavior-related parameters whose information is needed for precise pre-planning of NPs-assisted photothermal therapy [20]. This information can be related to the diverse biological and thermal behavior of different tissues during NPs-assisted PTT.

To pre-define the optimum setting for a thermal-based treatment method, in silico models and simulations are useful, as they allow for the investigation of the influence of the factors affecting the distribution of heat within the tumor and surrounding normal tissues. Some numerical- and simulation-based studies have been performed in the past on the laser-induced thermal therapy of cancer with the assistance of GNRs distributed within the tumor. Most of these studies have used finite element method (FEM)-based simulation software, such as COMSOL Multiphysics. As an example, Mooney and colleagues implemented a finite element model on COMSOL Multiphysics to simulate GNRs-enhanced laser therapy in mice and compared the results with experimental data [21]. Rossi et al. used the same software to evaluate the laser irradiation-related parameters for GNRs-based hyperthermia of cancer tissue. They proposed a model to design the correct laser settings with the goal of effective temperature induction within the tumor and avoiding thermal damage at the nanoscale level to the GNRs [22].

FEM-based simulation software makes possible the modeling of heat-transfer problems with the aim of performing a parameter-sensitivity analysis, and it is useful to handle complicated geometry, although it can be restricted by the rigid reliance on elements meshes [23]. Indeed, the result precision and computing time depend on the discretization of the domain (i.e., mesh size). The biggest weakness of this approach is related to the correct definition of the light propagation in biological media. In this environment, the propagation of photons can be described by numerically solving the diffusion approximation of the radiation transport equation, which is limited to systems where reduced scattering coefficients are much larger than their absorption coefficients. In this approximation, the medium that the light is propagated through is considered mostly a diffuse scattering medium with no distinction between absorption and scattering. This estimation is based on hypotheses that often present some limitations for the simulation of the real scenario in biological media [24]. By considering the biological tissue as a multilayer system with a non-negligible absorption coefficient, simulating the light propagation in biological tissues with the Monte Carlo method can provide more accurate results than the analytical method.

Among the Monte Carlo-based simulation codes, Geant4 [25,26,27] Application for Emission Tomography (GATE) is an open-source platform that has made possible the modeling of accurate optical photon transport in a tissue and is currently widely used for planning the radiotherapy and imaging related system. Currently, GATE supports simulations of Emission Tomography, Computed Tomography, Optical Imaging and Radiotherapy experiments. This platform allows for the obtaining of versatile and modular simulation toolkits, suitable for the design of new medical imaging devices, for the optimization of acquisition protocols and for the definition of new treatment protocols, such as the GNPs-mediated thermal therapy [28,29].

In the present work, we have designed the first GATE-based platform for GNRs-mediated PTT of pancreas tumor. Pancreatic cancer is an aggressive malignancy, which currently represents the fourth leading cause of cancer death in USA, and it is estimated to become the second leading cause of tumor-related death by 2030 [30]. Pancreas tumors can grow in the ducts, tail, or head of this organ, and are often inaccessible by traditional surgery for removal. For this reason, modern approaches including locoregional ablation [31] through percutaneous or echoendoscopic (EUS) guidance [32] have shown a significant potential. Although a few studies have been already carried out on the thermal ablation and hyperthermia of pancreas tumors [33,34,35], solutions for optimizing the technique in terms of treatment preplanning and selectivity are still at a premature stage. In the present work, we evaluate the thermal behavior of the pancreas when it is loaded with GNRs and compared the results with a model without GNRs. The pancreas model was generated from the CT image of the abdomen part of the human body, the pancreas phantom and the tumor were simulated with GATE software. The tumor was defined alone and loaded with GNRs in two separated programs. Based on the photon-deposited energy, we could calculate the temperature distribution within the pancreas. The temperature values predicted by the model have been compared with experimental data, in order to validate the results of the GATE simulations. The accuracy of the model in presence of GNRs was also verified by comparing the temperature increase obtained during laser irradiation of GNR suspensions in distilled water with the results of the simulations performed at the same settings. Lastly, we used the GATE-based platform to assess the influence of main procedural settings (laser power, laser applicator diameter) on the GNRs-mediated PTT of pancreas tumor.

## 2. Materials and Methods

GATE allows for simulating the absorption of optical photons by the healthy tissue and GNRs-loaded tissue, as well as the diffusion of the deposited photon energy (or heat) in biological tissues. In the following sections, we explain the definition of the geometry and the material used as the domain of our model, and the approaches used to compute the temperature increase within the tumor in the absence and presence of GNRs.

### 2.1. Simulation

#### 2.1.1. GATE: Geometry and Material Definition

The definition of geometry is a key step in designing the simulation because the radiation source, imaging device, object to be scanned, and detectors are described through the geometry definition [36,37]. Particles are then tracked through the components of the geometry. To perform a simulation with GATE, a definition of the “world” object is required. The “world” is the volume (box centered at the origin) that should be defined when starting a macro. Macro files contain an ordered collection of commands to define a geometry, to decide which particles to shoot as well as which energy to be considered for particles, to execute other commands and finally to shoot a particle. Macro can be executed from within the command interpreter in Gate [25].

Various physics phenomena, such as electromagnetic process (photoelectric effect, Compton and Rayleigh scattering, ionization, electromagnetic optic, etc.) and optical physics processes (bulk absorption, Mie scattering, wavelength shifting or fluorescence, etc.) as well as electromagnetic fields, can be defined and initialized. Different types of sources, each one is independent, such as ion source and simple particles (electron, positron, gamma, etc.) can be defined in the GATE simulation. Moreover, the activity of the given source can be defined by setting the parameter “setActivity” to the value that represents the rate at which number of particles are emitted and has the unit of Curie (Ci) or Becquerel (Bq). Intensity and the position of the source distribution can also be defined using specific commands [29].

The optical photons emitted by an illumination light (i.e., laser) are absorbed by the tissues through the optical absorption process. They can also be absorbed by a tissue that is loaded with a certain concentration of nanoparticles through the nano-absorption process whose additional details are discussed in the later section entitled “GNRs simulation”.

GATE models the heat diffusion in biological tissues by analytically solving the Pennes’ equation via Fourier transformations and convolution theorem [28]:(1)$$\frac{\partial T}{\partial t}=\frac{k}{\rho c}\left(\nabla^{2}T\right)+\frac{\rho_{b}c_{b}}{\rho c}w_{b}\;\left(T_{b}-\;T\right)+Q_{\mathrm{source}}\;\mathrm{with}\;T=T(x,\;y,\;z,\;t)$$
in which *k* (W/m∙K) is thermal conductivity, *ρ* (kg/m^3^) is density, and *c* (J/kg∙K) is the heat capacity of the tissue. Tissue thermal diffusivity (*α*) is defined by the term (*k*/*ρc*) and measures the rate of heat transfer in material from the hot end to the cold end [38]. The terms $\rho_{b}$, $c_{b}$, $w_{b}$ are the blood density, blood heat capacity and blood perfusion rate per volume unit, respectively. $T_{b}$ is the blood temperature that is assumed to be constant, and T is local tissue temperature which is a function of time and spatial position. The term $Q_{source}$ refers to the energy deposition by any external heat source (such as the metabolic heat production in tissues), respectively. By setting the $Q_{source}$ to 0 and taking the Fourier transform of spatial variable of Equation (1) an ordinary differential equation will be obtained:(2)$$\frac{\partial }{\partial t}ℱ\;\left(\;T\prime \right)+\left(\omega^{2}\alpha +Q_{b}\right)\;ℱ\left(\;T\;\prime \right)=0\;\mathrm{with}\;\alpha =\frac{k}{\rho c},\;Q_{b}=\frac{\rho_{b}c_{b}}{\rho c}w_{b}\;\mathrm{and}\;\;T\;\prime =T-T_{a}$$

α and $w_{b}$ are the tissue thermal diffusivity ($m^{2}/s)$ and the blood perfusion term, respectively, and ω is the Fourier transformation variable.

To solve the Equation (2) we can take the equivalent equation:(3)$$\frac{\partial }{\partial t}\left[e^{(\omega^{2}\alpha +Q_{b})t}\;ℱ\left(\;T\;\prime \right)\right]=0$$
and by using the partial integration with respect to t, the solution of Equation (3) yields:(4)$$ℱ\;\left(\;T\;\prime \right)=c\left(\omega \right){\;e}^{(\omega^{2}\alpha +Q_{b})t}$$
in which c(ω) is the Fourier transform of [T(*x*,*y*,*z*,0)-$T_{b}$], and it is expressed as c (ω) = ℱ ($T^{\prime }\left(x,y,z,0\right)$. The term T(*x*,*y*,*z*,0) is the initial temperature distribution of the tissue and is considered as initial condition. Lastly, the analytical solution for the Equation (1) is obtained by taking the inverse Fourier transform of Equation (4) and applying the convolution theorem for Fourier transformations:(5)T(x,y,z,t)=[T(x,y,z,0)− Ta]⊗1(4παt)32 e(x2+y2+z24αt)×e−Qbt+Tb

The blood perfusion term ($Q_{b}$) appears as an exponential function. In fact, the bioheat equation is solved by considering the initial condition (i.e., the 3-dimensional energy map) and a Gaussian laser beam with a standard deviation $s=\sqrt{2t\alpha }\;$ in which t is the irradiation time.

The Insight Segmentation and Registration Toolkit (ITK) [39] whose details are available in [28] is used to the implementation the heat diffusion in GATE. The standard deviation of the mean (i.e., voxel statistical uncertainty) was calculated at simulation time by considering the actual number of photons releasing their energy in the voxel (N_p_) and it scales as $\frac{1}{\sqrt{\mathrm{Np}\left(\mathrm{Np}-1\right)}}$ [40]. The simulations were carried out for $2\times 10^{6}\;$ proton histories that yielded lower than 0.1% statistical errors.

#### 2.1.2. Definition of the Pancreas Model

The current simulation was performed by modeling the laser irradiation in the pancreas phantom whose geometry was obtained from the segmented organ of an abdominal CT scan (Figure 1). In this regard, the pancreas was segmented from the CT scan file of the abdomen part of the human body using 3D Slicer platform [41]. The anonymized images were taken from a publicly available archive (https://www.cancerimagingarchive.net/, accessed on 23 October 2022). Clinical data stored in various image file formats can be read with GATE; moreover, users can access attenuation maps or emission data from the voxelized phantom. To import the patient data as a geometry to the GATE, the open-source software of XMedcon [42] was used to create a header (h33) and image (i33) files which are the accepted interfile format in GATE. The interfile consists of two files, one is “patient.h33” (the patient can be replaced with any name) which is an “ASCII” file with the header description of the image (sizes, spacing, origin and other information), and other is “patient.i33“ which has the pixels values as binary data for the image. GATE requires a text file (“AttenuationRange.dat”) for the description of material. The text file must provide a number of subdivisions, define intervals associated with each subdivision and attach them with a correlated material name. Through this file, the appropriate material can be assigned to each pixel with a certain value.

The 4 cm^3^ cubic phantom placed at origin (0, 0, 0) (in black color (Figure 1)) was considered as “world” then as a daughter, the pancreas image data was inserted within that by using the appropriate extension file (patient.h33) to create a pancreas phantom. The phantom was filled with the pancreas material with the density *ρ*=1040 kg/m^3^ by taking into consideration the real composition and the 3 cm^3^ tumor was defined within the head of the organ. The pancreas composition and density are from an ICRU Report [43].

The optical properties of the pancreas, such as refractive index, absorption and scattering coefficients, and absorption and scattering cross sections, were obtained and calculated using the data available in the literature [44,45,46,47,48,49,50,51,52]. These parameters are defined in a property table for the materials and are stored in a file (Materials.xml) separate from the material database (GateMaterial.db). This makes it easier to change the properties without having to change the material database. When GATE reads the material from the materials database, it also checks if the Materials.xml file contains a property table for this material. If so, this table is read in and coupled to the material.

The initial temperature of 26 °C was considered in our simulations, in order to replicate the environmental conditions of the benchmarking experiments reported in the Section 2.1.4 and Section 2.1.6 The thermal diffusivity of the pancreas was *D* = 0.14 mm^2^/s (Table 1), which was measured in our recent experimental study [53].

#### 2.1.3. Laser Source

A mono energy laser source with the wavelength of 808 nm generating $2\times 10^{6}$ optical photons per second (becquerel (Bq)) was defined at the center of the tumor (0, 0, 0) and the beam direction was set along the z-axis (Figure 1), perpendicular to the phantom surface at the xy-plane. Considering the typical size of laser applicators and power values used in clinical practice [32], two different laser beam diameters (i.e., 0.3 mm and 0.6 mm) and power values (i.e., 1.2 W and 2.1 W) were set. The power density was calculated by considering the number of generated photons per second by source and the diameter of the tip of the source.

#### 2.1.4. Temperature Distribution in Pancreas Phantom

To collect information during the simulation, such as energy deposit, number of particles created in a given volume, etc., “Actor” is needed as a tool which allows for an interaction with the simulation. Different actors such as “DoseActor”, “EnergySpectrumActor”, “EmCalculatorActor”, “ThermalActor”, etc., are the viable actors in GATE. Using a command of “attacheTo”, the actor is attached to the volume wherein the output of simulation supposed to be evaluated and obtained inside. For track and step levels, the actor is activated for step inside the volume and for tracks created in the volume. If no attach command is provided, then the actor is activated in any volume. Thermal actor records the optical photon deposited energy (photons absorbed by the tissue/material) in the volume which the actor is attached to. It also performs the diffusion of the deposited energy. The output file format is a 3D matrix (voxelized image with the format of .img/.hdr). We used the thermal actor to evaluate the absorbed energy in tumor. GATE provides a voxelized energy map (photon deposited energy in eV). To obtain the amount of temperature increment, the deposited energy is then converted into heat in an automated post-processing step [55]. The pancreas phantom was voxelized using the 0.25 mm^3^ cubes and the temperature was obtained by using simulated “detectors”, which were defined at a 2 mm horizontal distance (hd) from the tip of the source (parallel to the beam direction, (x,y,z=0, 2, 0)) and vertical distance (vd) from the tip of source (along the beam direction, (x,y,z=0, 0, 2)). The absorbed energy was then calculated by scoring the photon’s energy deposited in the voxels. As the GATE simulation output is a 3-dimensional matrix of the optical photon deposited energy (eV), the temperature increased inside the tumor without NPs was calculated following the approach in Qin publication [55].

To convert the energy absorbed by each voxel into heat, we need a conversion factor, F, which is calculated as:(6)$$F=\frac{1.6\times 10^{-19}}{m\;{\left(\mathrm{gr}\right)}_{\mathrm{voxel}}\times {\;C}_{p\;}\left(J\cdot g^{-1\;}\cdot {}^{\circ}C^{-1}\right)}$$
in which *Cp* is the heat capacity of the pancreas and *m* is the mass of each voxel, which is calculated by the multiplication of pancreas density and volume of each voxel (Table 1). The heat-induced within each voxel is then calculated by multiplying the value of the voxel by F:(7)Voxel (°C)=Voxel (eV)× F

We also considered a photon flux scaling factor to reproduce the experiment on pancreas tissue that we carried out for validating the simulated model in the post-processing step of heat calculation which provided a good comparison of both results. It can be measured using the Equation (4)
(8)$$\mathrm{photon}\;\mathrm{flux}\;\mathrm{scaling}\;\mathrm{factor}\;=\frac{P}{E_{0}\times 1.6\times 10^{-19}\times \pi \times {\left(\frac{d}{2}\right)}^{2}}$$
in which the P (W) is the power of the laser beam, E_0_ (eV) is the beam energy and d (m) is the laser source diameter. For instance, a 0.3 mm diameter-laser source with the wavelength of 808 nm (which corresponds to the beam energy (E_0_) of 1.5345 eV, generates around $12\times 10^{19}$ photons per second per ${\mathrm{cm}}^{2}$, when the power (P) is 2.1 W. So, we took $6\times 10^{13}$ as photon flux scaling factor in the post-processing step of heat calculation for the simulation in which we used Bq of $2\times 10^{6}$ (photon/s) with the same tip diameter of the laser.

##### Benchmark of the Simulation: Ex Vivo Experiments with Pancreas

The accuracy of the pancreas model, in absence of GNRs but in a scenario closer to a future clinical application, was verified by comparison with measured thermal distribution in ex vivo pancreas undergoing laser treatment. An ex vivo healthy porcine pancreas underwent a contact laser ablation with the power of 2.1 W for 120 s. A quartz optical fiber of 0.3 mm core diameter was used to guide an 808 nm continuous-wave laser beam (LuOcean Mini 4, Lumics, Berlin, Germany) on the tissue surface (Figure 2).

The temperature distribution was measured by employing a network of fiber optic sensors. Fiber Bragg grating (FBG) sensors were used for the scope. An FBG consists of a reflector resulting from a periodic modulation of the refraction index of an optical fiber core. FBG sensors offer important advantages such as electromagnetic immunity, high sensitivity, biocompatibility, and, thanks to their material, they are not prone to temperature overestimation. Additionally, they provide multi-point temperature measurements using a small-sized optical fiber embedding multiple FBG sensors, thus enabling the reconstruction of the temperature profile with accuracy for each sensor of 0.1 °C. Additional details on FBG sensors are reported in the previous papers of our group [12,56,57,58,59,60].

In this study, one FBG array containing 25 uniformly distributed FBG sensors with a length of 0.9 mm and a center-to-center distance of 1 mm was used. The FBG array was placed in parallel with respect to the laser applicator at a 2 mm distance (Figure 2b) to measure the temperature in proximity of the laser applicator. The laser applicator tip was aligned with center of FBG sensing portion (visible in the figure as irradiating fiber).

Reflected spectra from FBG array with a 100 Hz sampling rate were measured using Micron Optics si255 optical interrogator (Micron Optics, Atlanta, GA, USA). LabVIEW-based program was used to monitor in real-time the temperature profile and to save data (Figure 2a). Afterwards, the data were analyzed using Matlab R2018a^®^ software. The experiments were repeated three times with the same settings and on different locations of the organ, to avoid overlapping thermal treatments. Temperature results are calculated as the average value of the three repetitions, with the associated standard deviation.

#### 2.1.5. GNRs Simulation

In our simulation, 12.5 µg/mL GNRs with a diameter of 10 nm and a length of 41 nm (aspect ratio = 4.1) were considered [11] and uniformly defined within the tumor with a volume fraction of around $6\times 10^{-7}$. The absorption and scattering cross sections of GNRs in the mentioned size were calculated by using the equation derived from Gans theory, which is discussed later. For the tumor loaded with GNRs, the tissue materials were re-calculated by considering the percentage of GNRs presence and thus the density of the pancreas tumor embedding GNRs was increased compared to the one only associated with the pancreas tumor tissue. The absorption length of GNRs within the pancreas was measured to be around 0.9 mm. The conversion of absorbed energy recorded by the detector to temperature was carried out by using the approach introduced by Cuplov et colleagues [28].

Moreover, to simulate GNRs within the pancreas, we used the approach proposed by Cuplov et al., which relies on the definition of a material with a property called NanoAbsorptionLength (La) and an optical photon physics process called NanoAbsorption. In GATE, each optical photon is transported following a step length which is randomly sampled using the interaction length (also known as mean free path) of each physics phenomenon process associated with the optical photon such as Rayleigh or Mie scattering, fluorescence, and physics processes at the interface between two media, including the nanoabsorption process.

In GATE simulation, the interaction length of the optical photon interaction with the nanoparticle-infused medium is defined by the parameter of La. It represents the average distance of an optical photon travelling in the nanoparticle-infused medium before being absorbed. The inverse of the absorption length is referred to as the absorption coefficient (µa). This coefficient is a function of the density of NPs in the medium (N in the number of NPs/m^3^) and the NPs absorption cross-section area ($C_{\mathrm{abs}}$ in m^2^):(9)$$\mu_{a}=N\times C_{\mathrm{abs}}$$

The particle absorption cross section is calculated by [61]:(10) 〈Ca〉=4πkIm〈α〉=4πk3 Im ∑i=1,2,3αi

Polarization tensor of a homogeneous ellipsoid, $\alpha_{i}$, with a dielectric permittivity function of ε(λ), in a medium with a dielectric permittivity function $\varepsilon_{m}\;\left(\lambda \right)$ is calculated through the following equation:(11)$${\;\alpha }_{i}=4{\pi D}^{2}l\;\left(\frac{\varepsilon -\varepsilon_{m}}{3P_{i}\left(\varepsilon -\varepsilon_{m}\right)-3\varepsilon_{m}}\right)$$
where in D and l are the diameter and length of GNRs respectively and i = 1, 2, 3, which represents the number of axes. $P_{i}$ is the geometric factor that is given by the equation:(12)$$p_{1}=\frac{1-x^{2}}{x^{2}}\;\left[\frac{1}{2x}\ln\left(\frac{1+x}{1-x}\right)-1\right]$$
where $x=\sqrt{1-{\left(\frac{D}{l}\right)}^{2}}$ and $p_{2}={\;p}_{3}=\frac{1-p_{1}}{2}$.

ε and $\varepsilon_{m}$ are the dielectric functions of GNRs and medium embedding the GNRs (pancreas), respectively.

Optical constants of small particles with radii less than 10 nm differ from the bulk values due to the limitation of an electron mean free path. When the size of the metallic particles becomes smaller and reaches the few nanometers range, the optical properties are modified by the appearance of surface plasmon, and its behavior results completely different from that of bulk metal. The intrinsic size effects are related to the damping of the electron oscillations. When surface plasmons are excited, the electrons are damped in their movement by the scattering into the ionic cores and into the surface.

The size influence on the optical constants of small particles with the radius of a is convenient to express in terms of the particle dielectric permeability $\;\varepsilon_{a}$
(13)$$\varepsilon_{a}=\varepsilon_{\mathrm{bulk}}+\Delta \varepsilon \left(a\right)$$

Δε(a) is the correction accounting for the size-dependence contribution to the electron mean free path, which is as a difference of two Drude terms (Equation (10)): [62,63,64]
(14)$$\Delta \varepsilon \left(a\right)=\frac{\omega_{p}^{2}}{\omega \left(\omega +\frac{i}{\tau }\right)}-\frac{\omega_{p}^{2}}{\omega \left(\omega +\frac{i}{\tau_{a}}\right)}$$
(15)$$\varepsilon_{D_{i}}\left(\omega \right)={\;\varepsilon }_{0}-\frac{\omega_{p}^{2}}{\omega^{2}+{\;\gamma }_{\mathrm{bulk}}^{2}}+i\frac{\gamma_{\mathrm{bulk}}\omega_{p\;}^{2\;}}{\omega \left(\omega^{2}+\gamma_{\mathrm{bulk}}^{2}\;\right)}$$
where in ω_p_ is bulk plasma frequency ($\sqrt{\frac{ne^{2}}{\varepsilon_{0}m_{e}}}$) and $\gamma_{\mathrm{bulk}}$ is damping constant proportional to the reciprocal of the mean free time between electron collisions in metal and describes the damping of electron oscillation due to the scattering of these oscillating electrons with the ionic cores. It is size-independent and determined as the ratio of Fermi velocity ($v_{F}:$ velocity of conduction electron) to the electron mean free path of *l* (42 nm for gold [65]).

The relaxation time of τ ($\tau =\frac{1}{\gamma })$ is related to the bulk and modified as [66]:(16)$$\frac{1}{\tau_{a}}=\frac{1}{\tau }+\frac{{\mathrm{gv}}_{F}}{a}$$

The second term corresponds to the scattering of oscillating electrons with the particle surface. When the size of particles decreased to a nanometer, electrons in a certain shell close to the surface will scatter this surface when they oscillate. The proportionality coefficient, g, is a material-dependent constant and was taken equal to 1 according to references [67,68]. The equivalent radius of GNRs was calculated using the s-cylinder model introduced by Alekseeva et al. [61] (Figure 3):(17)$$R_{\mathrm{ev}}={\;b}_{3}s\;\sqrt{1+\frac{3a}{2b}},\;e=\frac{a+b}{b}$$

In this model, the rotational symmetry is considered and two parameters of thickness 2b and length 2(a + b) were taken to calculate this parameter.

The optical parameters of the GNR-loaded pancreas tumor were calculated following the different approaches in the literature. The information about the pancreas and GNR-loaded tumor properties used in our model is available in Table 1. According to our recent experiment [20] performed on GNR-loaded agar phantom, the diffusivity of the phantom loaded with GNRs increased around 22% at a temperature of around 62 °C compared to the one at room temperature. So, in this study, we considered the average value of diffusivity from room temperature to be 62 °C [20]. To estimate the refractive index (RI) of pancreas tumor loaded with GNRs, the mole fraction of pancreas tumor alone and with GNR was calculated separately. Then, the mole fraction of each component (pancreas and gold) was multiplied by the RI of that component (Pancreas RI = 1.394 and gold RI = 0.155) and by the sum of all products the RI of GNR-loaded tumor was estimated [69].

#### 2.1.6. Temperature Distribution in GNR-Loaded Pancreas Phantom

Following the approach in the review by Qin et al. [55] that focused on the thermophysical, and biological responses of media heated by laser-activated GNPs, the heat generation within the GNPs-loaded tumor with a concentration N (number of NPs/m^3^) of GNPs, is given by:(18)$$Q_{\mathrm{heat}}=N\times Q_{\mathrm{NP}}=N\times C_{\mathrm{abs}}\times I$$
where $Q_{NP}$ is the heat generated by a single nanoparticle, $C_{abs}$ is the nanoparticle absorption cross-section area and I is the local laser irradiance (in W∙m^2^) within the nanoparticle-infused tissue. Here $C_{abs}$ for the pancreas tumor loaded with 12.5 µg/mL GNRs with aspect ratio of 4.1 was calculated using Equations (10)–(15) and was measured around 476 nm^2^.

The temperature increase at the center of the tumor is given by the following equation:(19)$$\Delta T=\frac{N\times R^{2}\times {\;C}_{\mathrm{abs}}\times I}{2k}$$
where R is the tumor radius and k is the thermal conductivity of the nanoparticle-infused tissue.

The 3-dimensional energy map is voxelized (the voxel size is a parameter of the simulation) and therefore ΔT can be scaled to reflect the heat increase in temperature per tumor volume unit (i.e., per voxel). In that case, R is replaced by the voxel half-size, and I become the light irradiance per voxel (I_voxel_):(20)$$I_{\mathrm{voxel}}=\mathrm{voxel}\;\times \frac{1.6\times 10^{-19}}{t_{\mathrm{laser}}\times {\;\mathrm{area}}_{\mathrm{voxel}}}$$
where “voxel” is the GATE simulation energy map voxel value in electron volts, the laser is the light (i.e., laser) illumination duration in seconds, area voxel is the voxel surface in m^2,^ and $1.6\times 10^{-19}\;$ is the conversion factor between electronvolts and joules. The photon flux scaling factor was also considered to calculate the absolute generated heat within the GNR-loaded tumor.

##### Benchmark of the GATE Simulations: Experiment on GNRs Suspension

The accuracy of the pancreas model in presence of GNRs was verified by comparing the temperature increase obtained during laser irradiation of GNR suspensions in distilled water with the results of the simulations performed at the same settings. Due to the lack of blood circulation in the ex vivo study, the distribution of GNRs cannot be achieved properly in the direct injection of particles to the pancreas. Depending on the type of tissue, clinical applications as well as desired optical properties, there are various phantom matrices such as an aqueous suspensions including water, gelatin or agar-based matrix, and silicone [70]. Here we used water, as it allows for the control of the distribution of the NPs to be as homogeneous as possible.

To this end, a drop of 0.1 mL of the suspension was placed inside a well of 24-multiwell cell culture plate and irradiated with the continuous-wave 808 nm laser beam (LuOcean Mini 4, Lumics, Berlin, Germany) for an exposure time of 30 s. The laser beam was delivered in a contactless modality by means of an optical collimator (Figure 4). Experiments were conducted at two different laser power (i.e., 1.2 W and 2.1 W). Moreover, 11-mercaptoundecyltrimethylammonnium bromide (MUTAB)-coated GNRs (Nanopartz™, Inc., Loveland, CO, USA) with a peak absorption at 808 nm, a diameter of 10 nm and a length of 41 nm, thus an aspect ratio of 4.1, were employed. The laser-induced temperature increase was measured in the distilled water alone (i.e., control) and in GNRs suspensions in distilled water (0.625 mg/mL GNRs concentration) using an infrared thermographic camera (FLIR System, T540 with 464 × 348 pixels spatial resolution, ±2 °C accuracy). The experiment was repeated three times.

The same setup was simulated with GATE by defining a 0.1 mL water drop inside the well with the same size and the temperature increase was obtained at the surface of the water in the presence and absence of GNRs (with the same concentration utilized in the experiments) for 30 s irradiation at 808 nm. The optical properties of water and GNRs were calculated following the methods described in the previous sections, and by using data derived from the literature [48,71,72,73]. The heat generation within the water (with and without GNRs) was simulated by the aforementioned post-processing method and the comparison was made with the results obtained in the experiments.

## 3. Results

### 3.1. Validation of the GATE on the Pancreas Model and on the GNRs-Assisted Temperature Increase within the Tumor Phantom

In this section, we report the results on the validation of the model, based on the experiments performed in ex vivo pancreas and in water with and without GNRs. Then, the heat distribution computed with the implemented GATE-based model was compared for the pancreas alone and pancreas with NPs. Results are compared in terms of ΔT, which represents the temperature change resulting in the tissue after the laser irradiation, with and without GNRs.

#### 3.1.1. Validation of the Pancreas Model

We compared the experiments on ex vivo pancreas and simulations in Figure 5. The temperature profile measured in the experimental step (in blue) is associated with the maximum temperature at 2 mm from the laser applicator estimated during 120 s of laser irradiation. On the other hand, the simulation (in orange) is showing the temperature profile extracted at hd = 2 mm from the tip of laser in parallel to the beam irradiated for 120 s. Specifically, the temperature in the voxel at 2 mm distance from the laser applicator was chosen for the comparison.

#### 3.1.2. Benchmark of the GATE Simulations for GNRs-Loaded Pancreas Tumor Model

Figure 6 shows the comparison of temperature enhancement induced by 30 s laser irradiated inside the water alone and water loaded with 0.625 mg/mL GNRs for powers of 1.2 and 2.1 W and between the data obtained from the experiment (in blue) and simulation (in orange). The graph reports the maximum ΔT within the phantom. The relative difference which is the systematic error can show the accuracy and was measured in %:(21)$$\mathrm{Relative}\;\mathrm{error}\;\%=\left|\left(\frac{Experiment-Simulation}{Experiment}\right)\times 100\right|$$

For the water alone, the relative error was 11.76% and 3.45% for 1.2 W and 2.1 W, respectively. When the water was loaded with GNRs this parameter was measured as 5.28% and 13.48% for the powers of 1.2 W and 2.1 W, respectively.

### 3.2. Simulation Results: Temperature Distribution in the Pancreas, in the Presence and Absence of GNRs

The maximum irradiation time was set to 120 s; however, in some cases, the temperature of the tumor in the presence of GNRs increased to more than around 100 °C for a shorter irradiation time. For instance, laser irradiation of GNR-loaded pancreas with the power of 2.1 W caused the temperature increasement of around 200 °C just after ~85 s. Therefore, to compare the results, we selected a maximum temperature threshold of 155 °C and only the values lower than this threshold were included in the analysis, as we consider that, for a real application, temperature values >150 °C will create ablation of the medium and dangerous carbonization [74].

The maximum temperature increase at 2 mm-hd within the tumor in the presence and absence of 12.5 µg/mL GNRs were compared in Figure 7 when the tumor is exposed to a laser beam with two core diameters of 0.3 and 0.6 mm and power of 2.1 W. Figure 8 shows the comparison of temperature increase within the pancreas tumor with and without GNRs for 1.2 W and 2.1 W, at 2 mm-hd from the source. The core diameter of the source was 0.3 mm.

Figure 9 shows the temperature distribution within the tumor at 30 s irradiation for 2.1 W along one dimension (*x*-axis). Four different distances (z=0.5, 1, 2 and 2.5 mm) along the beam direction from the laser tip were measured and compared for the pancreas tumor with and without NPs.

To better visualize the temperature distribution within the tumor in the presence and absence of GNRs, we obtained the temperature increase by defining the detector at a 2 mm distance from the tip of the source and plotted the temperature distribution in the plane perpendicular to the laser beam direction (*xy*-plane). Figure 10 shows the results of this comparison after 30 s laser irradiation with a power of 2.1 W.

## 4. Discussion

The comparison between the experimental and simulated results for the irradiation of pancreas tissue shows a good similarity (Figure 5). A maximum difference of around 4.5 °C in the value of enhanced temperature was found (Figure 5) between the experiment and simulation with an overestimation of temperature in the simulation results. The difference between the experiment and simulation was reported with the Re ranges from around 10% to 19%. While the absolute error (experiment–simulation) is around 4.5 °C for irradiation time above 50 s, it is expected that the larger RE is associated with the smaller accepted value (experiment). Thirty-second irradiation of water in the presence and absence of GNRs also showed similar results, with a minimum and maximum relative error of around 3.5% and 13.5%, respectively, between the experiment and simulation (Figure 6). The difference between the simulation and experimental results can be explained by modification of material properties (e.g., changes in the thermal conductivity and diffusivity, occurrence of coagulation) which occur in the tissue and water during the laser irradiation [53,75,76,77]. Moreover, the presence of a high thermal gradient [56] can affect the temperature sensor accuracy [78,79].

According to the results represented in Figure 7, increasing the diameter of the tip of the laser source causes a decrease in the temperature enhancement in the GNRs-loaded pancreas during the irradiation, since the power density decreases. For instance, 10 s of irradiation using the laser source with diameters of 0.3 and 0.6 mm induced a temperature increase of 31 °C and 39 °C in the GNRs-infused pancreas, respectively. While ablation progresses, instead, the temperature increase for the two laser sources is different. At 40 s after irradiation, the temperature increase reached around 60 °C, for a diameter of 0.6 mm, whereas it reached around 100 °C for a 0.3 mm source at the same irradiation time. These results confirm that, in addition to factors such as GNRs size, concentration, and kind of particles, also laser parameters, such as the position and the diameter of the source, should be optimized according to the effect of GNRs on temperature enhancement.

Accordingly, we also compared the effect of the power increase on the thermal effect induced within the tissue in the presence and absence of GNRs (Figure 8). It is expected that the decrease in the power for a certain diameter of laser source induced higher temperature within the tumor. However, by comparing the tissue with and without GNRs, the results showed that the presence of NPs has a higher impact on the temperature increase than the power set. The difference between the results for the two power values increases with time (during PTT) for both the tissue alone and the tissue with GNRs. For instance, at 10 s, raising the power caused around 5 °C and 17 °C temperature increases within the tissue without and with NPs, respectively. These raisings in the temperature enhancement reach 64 °C at 70 s after irradiation when the tissue has GNRs, while it reached 13 °C within the tissue alone, at the same irradiation time. This is an exact explanation for the aforementioned point that emphasizes the importance of the precise evaluation of all the parameters for optimizing the setup for the preplanning of cancer thermal therapy.

Temperature enhancement within the tissue along the beam direction and along the x-axis became more localized by closing to the tip of laser, however, this localization was much more significant when the tissue was loaded with GNRs (Figure 9). At 2 mm-distance from the source (Figure 9, yellow lines) and by moving far from the source of 1 cm, in the xy-plane, the temperature increase drops to <1 °C when the pancreas tumor was loaded with GNRs (Figure 9D, purple line), while it drops to ΔT=6 °C from the at the same place, but in absence of GNRs (Figure 9C, yellow line).

These results indicate that for the same power emitted from a source with a certain wavelength, each second generates a certain number of photons whose energy is inversely proportional to the wavelength, E=hc/λ, h is Planck constant, c is the speed of light, and λ is the wavelength of the photon. The higher portion of irradiated photons’ energy is absorbed by GNRs and cause localized heat induction close to the source. So, in the usage of GNRs-enhanced thermal therapy, the more precise and homogenized distribution of GNRs within the tumor leads to a more localized heat induction in cancerous cells. However, to induce the more homogenized temperature increment within the whole region of the tumor, in addition to the distribution of an optimum number of GNRs that should be considered, the shaping and irradiation profile for the laser source are also important. This is what causes a difference in the distribution of energy absorbed by cancerous cells when the source is used externally or internally. It means that if the beam at a certain time can reach a bigger area of GNRs-loaded tumor assuming that the particles are homogeneously distributed within it, a temperature increase occurs in the whole tumor and causes treatment time to decrease.

The effect of the different absorption of photons caused by the presence of GNRs in the tissue is confirmed in the Figure 10. Indeed, the temperature increase within the tumor without GNRs is more widely distributed in the space while the presence of GNRs causes a higher portion of the heat to be absorbed by the tissue immediately surrounding the source. Since the local absorption is higher with GNRs (and in the direction of the laser beam), fewer photons penetrate the tissue in all the other directions.

While it is clearly confirmed that the presence of GNRs within the tumor can induce a required irradiation dose to the tumor in a shorter time, the effects of some factors such as power density, the size of the laser source, number of GNRs and the position of source relative to the tumor are important and should be considered.

In addition to PTT, the GATE software has the ability of modeling therapy protocol for radiation therapy as well as modeling the multimodal imaging (PET, CT, SPECT, optical imaging) [25]. So, this open-source simulation platform has the advantage of being used in designing, optimizing, and assessing the efficacy of theranostic protocols. Precise simulation for pre-planning of NP-mediated hyperthermia for different kinds of tumor and NPs require more information of thermal parameters. For instance, in the simulation of the heat diffusion the tissue, the thermal diffusivity plays an important role. In GATE, this parameter is set to a value that best represents the diffusivity of the mixture of a tumor surrounded by healthy tissues. This parameter is not easily available for different wavelengths and organs in the literatures. So, reliable simulation results with GATE for NPs-mediated hyperthermia of tumors require precise data and information for thermal parameters of different tissues and NPs.

Regarding the kind, size and concentration of NPs, laser beam irradiation of tumor in the presence of NPs can lead the temperature increase to above 100 °C, where evaporation occurs. So the heat-transfer model should be further improved to consider some phenomena, such as the water evaporation and the temperature-dependence of tissue properties [38].

Instead of attempting time-consuming trial-and-errors experiments, the Monte Carlo simulation software allows for the evaluation of the effect of different NPs distributions on the temperature distribution, in order to tailor the optimal distribution for each specific application. However, the real conditions are more complex, and NPs are surrounded by various compounds within a tissue that could cause disparities in the elicited responses. For instance, the absorption and scattering spectra of nanoparticles in water can be obtained by simulation but when the NPs are placed within the tissue it would be much more complicated. So, modelling the light propagation in biological tissues when they are loaded with nanoparticle still constitutes a major challenge, and a reliable method requires further study and investigation.

Additionally, the distribution of NPs in the simulation is generally considered to be homogeneous, while in the real-life condition they can accumulated within different parts of tissue. So, to achieve an increase in the accuracy of elicited outcomes, the coupling effects of clustered nanoparticles close to each other have to be considered.

## 5. Conclusions

This work presents the first GATE Monte Carlo-based platform for PTT in the pancreas. This platform allows for the modeling of the heat transfer in the organ and tumor embedding GNRs. Moreover, this code makes it possible to reproduce the experimental scenario and simulate the GNRs-mediated NIR thermal therapy with good accuracy. This work is quick to point out that a precise simulation of laser therapy for the pancreas with different NPs leads to a better evaluation of the optimum setting for pre-clinical and clinical treatment. Moreover, given that there are still some concerns in the clinical usage of NPs in cancer treatment, measuring the thermal parameters of the pancreas when it is loaded with NPs can be useful for precise pre-planning of treatment.

## Figures and Tables

**Figure 1 cancers-14-05686-f001:**
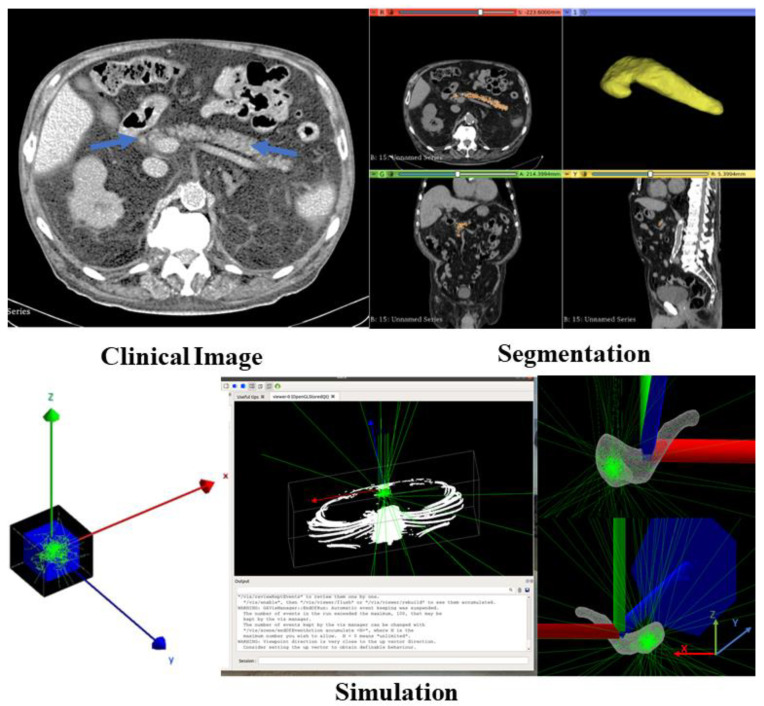
Segmentation of pancreas from the CT scan of abdomen part (clinical image) and simulation the pancreas phantom in GATE software.

**Figure 2 cancers-14-05686-f002:**
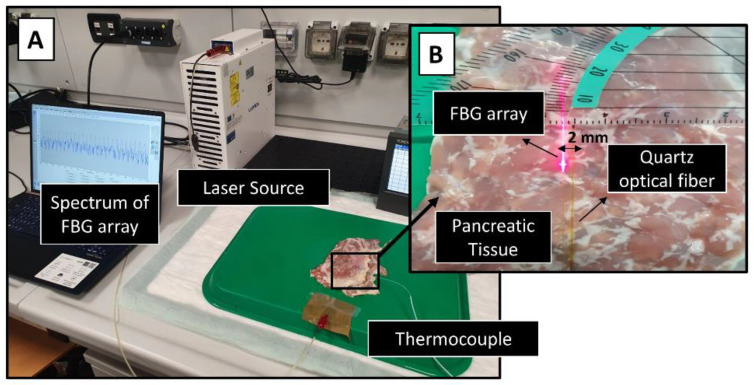
(**A**) Set up for the measurement of the temperature enhancement in ex vivo pancreas tissue during 120 s laser irradiation. (**B**) Magnification of the organ with the optical fiber placed in the middle part of that.

**Figure 3 cancers-14-05686-f003:**
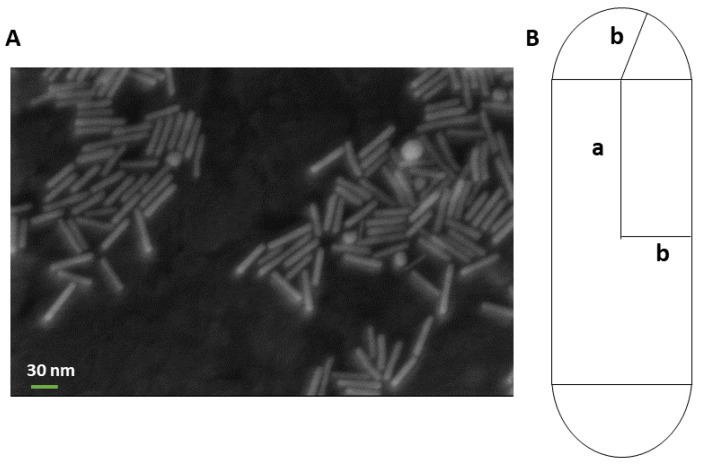
(**A**) TEM image of GNRs and (**B**) geometric model of s-cylinder [61].

**Figure 4 cancers-14-05686-f004:**
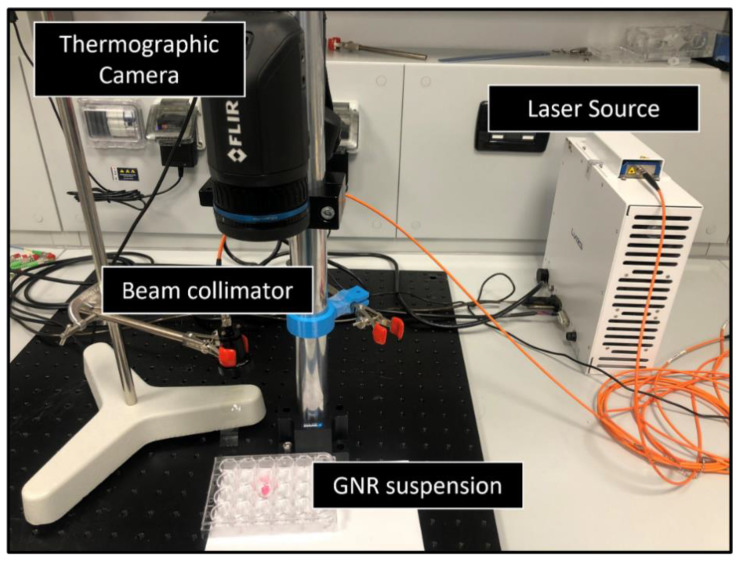
Experimental setup utilized for the near-infrared irradiation of GNR suspensions: the attained temperature results are used for the benchmark of the implemented simulation model.

**Figure 5 cancers-14-05686-f005:**
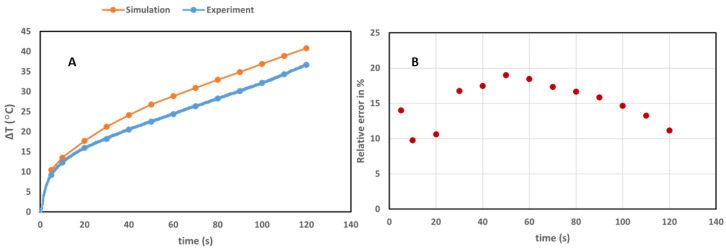
(**A**) Temperature increases at 2 mm distance from the tip of the laser applicator (diameter of 0.3 mm) guiding 808 nm laser light for the ex vivo (blue curve) and simulated (orange curve) pancreas, at a power of 2.1 W. (**B**) Relative error between the experimental data and simulation.

**Figure 6 cancers-14-05686-f006:**
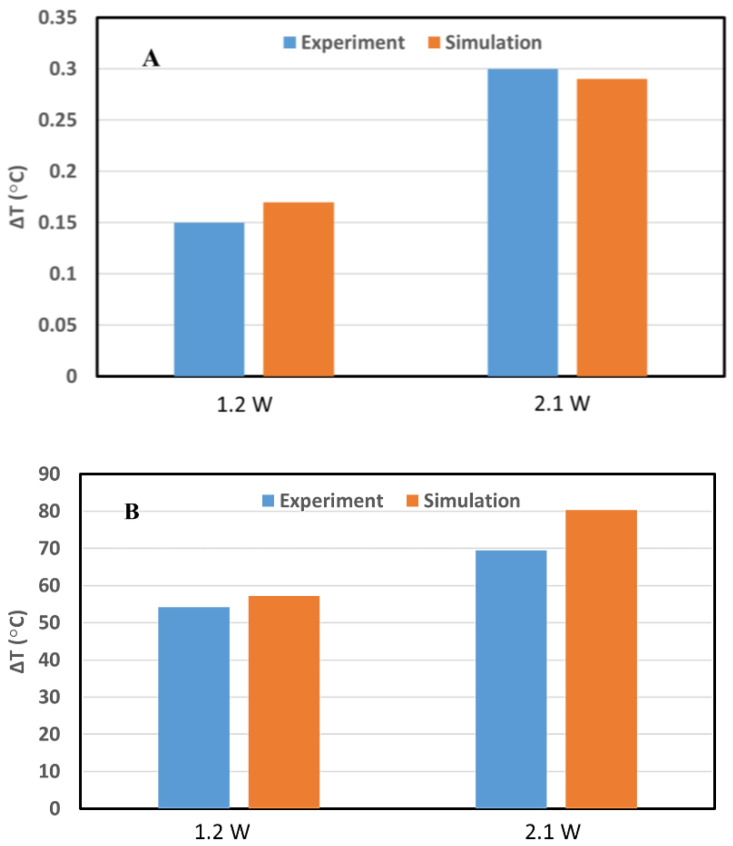
Comparison between simulation and experiment of the temperature increment during the 30 s laser irradiation in (**A**) water alone and (**B**) GNR suspensions for the laser powers of 1.2 and 2.1 W. For the powers of 1.2 W and 2.1 W, relative error is 11.76% and 3.45%, respectively, for water alone and is 5.28% and 13.48%, respectively, for GNR suspension.

**Figure 7 cancers-14-05686-f007:**
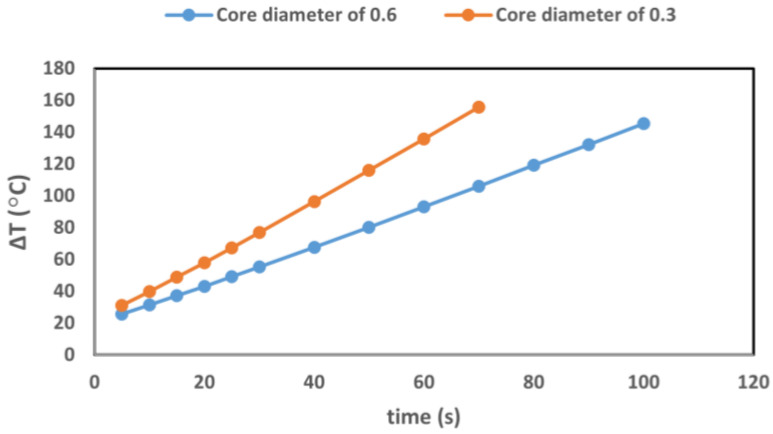
Temperature increases in the GNRs-loaded tumor when exposed to laser irradiation, at a power of 2.1 W with two laser applicator diameters: 0.3 mm (orange) and 0.6 mm (blue).

**Figure 8 cancers-14-05686-f008:**
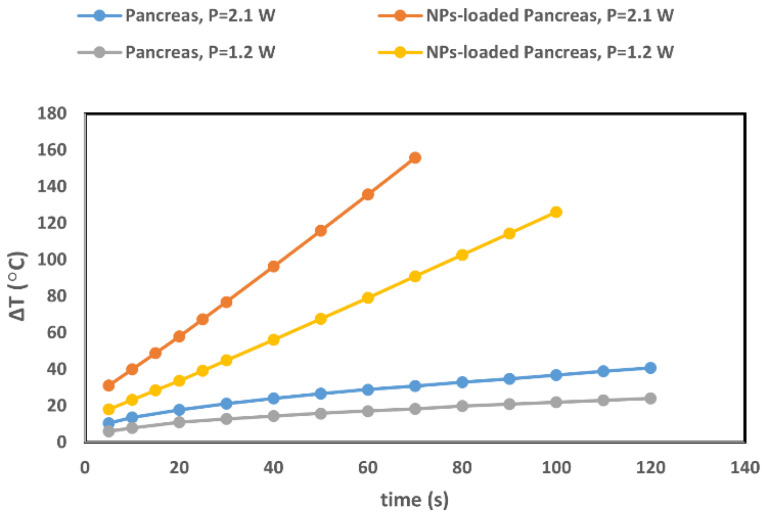
Temperature increases at 2 mm-hd from the light source in the pancreas tumor in the absence (gray and blue curves) and in presence of GNRs (yellow and orange curves), when exposed to laser irradiation, at two power values (1.2 W and 2.1 W) with laser applicator diameter of 0.3 mm.

**Figure 9 cancers-14-05686-f009:**
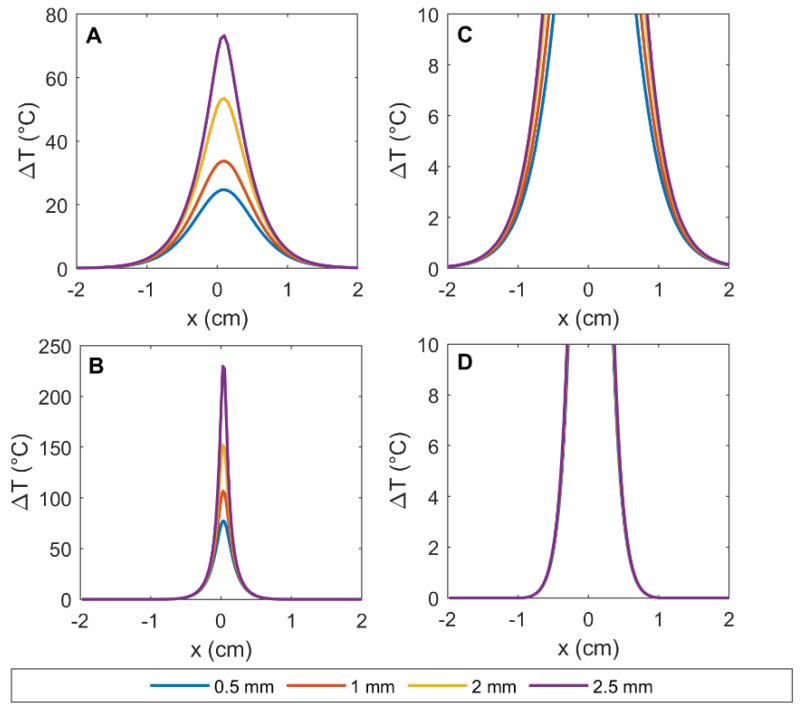
Temperature distribution along the x-axis within the tumor at 4 different points with the distance of 0.5, 1, 2, and 2.5 mm from the tip of the source and placed in the direction of laser beam and at 30 s after irradiation at P = 2.1 W, for (**A**) pancreas and (**B**) pancreas loaded with 12.5 µg/mL GNRs. A zoom of the curves in the ΔT range 0–10 °C is reported for (**C**) pancreas and (**D**) pancreas loaded with 12.5 µg/mL GNRs.

**Figure 10 cancers-14-05686-f010:**
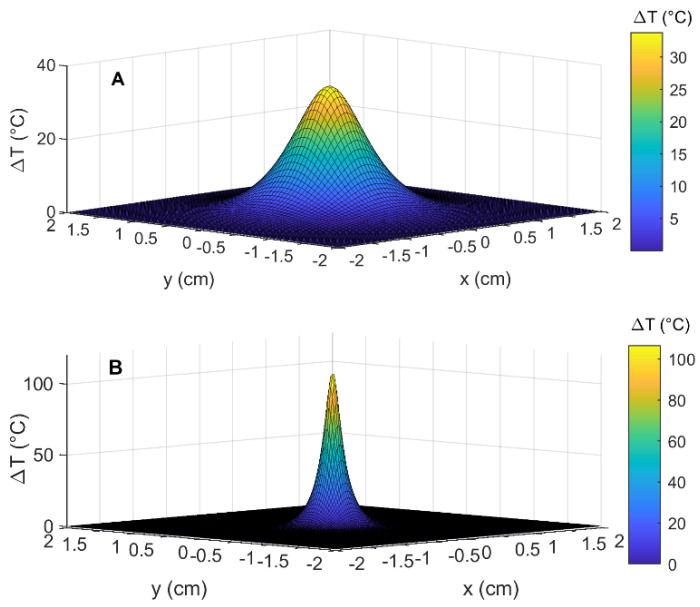
Temperature distribution in the *xy*-plane within the tumor at 2 mm distance from the tip of the laser source and placed in the direction of laser beam and at 30 s after irradiation for (**A**) pancreas and (**B**) pancreas loaded with 12.5 µg/mL GNRs. A laser power of 2.1 W was used to obtain these results.

**Table 1 cancers-14-05686-t001:** Properties of the pancreas tissue used in our model.

Parameters	Symbols	Values for Pancreas	Values for GNR-Loaded Tumor
Percent by weight element composition [43]	H	1.06	1.059
	C	1.69	1.68
	N	0.22	0.219
	O	6.94	6.939
	Na	0.02	0.019
	P	0.02	0.019
	S	0.01	0.0099
	C	0.02	0.019
	K	0.02	0.019
	Au	/	0.0012
Density [43]	ρ (kg/m^3^)	1040	1,040,012
Thermal diffusivity [20,53]	D (mm^2^/s)	0.14	0.16
Heat Capacity [20,53]	Cp (mJ/kg∙K)	3.5546	3.6057
Refractive index [52]	RI	1.394	1.3939
Absorption coefficient [51]	µ_a_ (1/cm)	0.0388	1.6850
Scattering coefficient [51]	π_s_ (1/cm)	196.8	0.02740
Permittivity [50,54]	$\varepsilon_{1}+\varepsilon_{2}i$	1.767 + 0i	24.7180 + 3.6581i

## Data Availability

Data are made available by the authors upon request.

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
