# Peer review of "Gold Nanoparticles-Mediated Photothermal Therapy of Pancreas Using GATE: A New Simulation Platform"

_cancers, 2022, doi:10.3390/cancers14225686_

Round 1

Reviewer 1 Report

The manuscript entitled "Gold nanoparticles-mediated photothermal therapy of pancreas using GATE: a new simulation platform" written by Somayeh Asadi Leonardo Bianchi et. However, as I read the manuscript, I found a few issues that need to be addressed before the paper meets the quality standard for publication.

1.      The distinction between abstracts and introductions must be made clear. The abstract must be modified. Please rearrange the abstract to clear your aim and the novelty. The abstracts of the article should be as simple and clear as possible, and the content of the whole article should be summarized comprehensively and accurately.

2.      Part 2: Materials and Methods. Suggest authors describe it in more detail. Materials and methods information which will affect their behavior in the subsequent work should be given. Also, details of instrumentation used for should be included.

3.      Line 518-521," Thirty-second irradiation of water in the presence and absence of GNRs in both the experiment and simulation studies showed similar results with a minimum and maximum relative error of 3.5 % and 13.5 %, respectively, between the experiment and simulation (Fig 5)." Relative error cannot be seen in Fig 5.

4.      There are some errors in the format of temperature units in the manuscript, please check and correct.

5.       Line 528-531, " It should be noted that factors like errors in sensor positioning due to high thermal gradient, and modification of the water and tissue during the irradiation (e.g., changes in the thermal and optical properties like thermal conductivity and diffusivity, occurrence of coagulation) can be among the causes of the difference between the simulation and experimental results. " Author should reframe the sentence to make it clearer to understand. And give the suitable reference for the same.

6.      Many sections in the manuscript are garbled. Please check the full text.

Author Response

We appreciate the reviewer for his/her thoughtful evaluation of this manuscript and constructive comments. We have carefully addressed all the comments, aiming at improving the structure and readability of our manuscript. All the changes have been highlighted in yellow color.

Manuscript # cancers-2020025

Reviewer’s Comments and Replies

Reviewer 1

  • The distinction between abstracts and introductions must be made clear. The abstract must be modified. Please rearrange the abstract to clear your aim and the novelty. The abstracts of the article should be as simple and clear as possible, and the content of the whole article should be summarized comprehensively and accurately.

Reply: Thank the Referee for reading the manuscript carefully and for his/her comments. The abstract was revised, and the content of manuscript was summarized in the last paragraph.

  • Part 2: Materials and Methods. Suggest authors describe it in more detail. Materials and methods information which will affect their behaviour in the subsequent work should be given. Also, details of instrumentation used for should be included.

Reply: Thanks to the Referee for this useful suggestion. In this regard the section of material and method was revised, some paragraphs were added, and some sections were reordered to better clarify this section.

  • Line 518-521," Thirty-second irradiation of water in the presence and absence of GNRs in both the experiment and simulation studies showed similar results with a minimum and maximum relative error of 3.5 % and 13.5 %, respectively, between the experiment and simulation (Fig 5)." Relative error cannot be seen in Fig 5.

Reply: We thank the Referee for reading the paper carefully. The relative errors were mentioned in the text before the figure, but this part was revised and the related information was added to the caption of figure (highlighted in yellow colour).  

  • There are some errors in the format of temperature units in the manuscript, please check and correct.

Reply: We thank the Reviewer for this observation. We have checked and corrected the typos.

  • Line 528-531, " It should be noted that factors like errors in sensor positioning due to high thermal gradient, and modification of the water and tissue during the irradiation (e.g., changes in the thermal and optical properties like thermal conductivity and diffusivity, occurrence of coagulation) can be among the causes of the difference between the simulation and experimental results. " Author should reframe the sentence to make it clearer to understand. And give the suitable reference for the same.

Reply:  Thank the referee for this comment. This paragraph was revised as follows, and some references were added to that:

“The difference between the simulation and experimental results can be explained by modification of material properties (e.g., changes in the thermal conductivity and diffusivity, occurrence of coagulation) which occur in the tissue and water during the laser irradiation [53], [75]–[77]. Also, the presence of high thermal gradient [55] can affect the temperature sensor accuracy [78], [79].”

  • Many sections in the manuscript are garbled. Please check the full text ".

Reply – We are thankful for this comment. The manuscript was deeply revised in this regard by adding some more explanations in different sections and reordering and simplifying some paragraphs, sections, and subsections. 

Reviewer 2 Report

The article by Asadi et al. is devoted to the application of the Monte Carlo-

based code developed with GATE for gold nanorods (GNRs)-assisted

therapy of a pancreas tumor. Benchmark tests demonstrated the

possibility of accurate simulating of NPs-assisted thermal therapy and

reproducing the experimental data. The output of simulated GNR-

mediated hyperthermia emphasized the importance of the precise

evaluation of all the parameters for optimizing the preplanning of cancer

thermal therapy. The article is full of mathematical calculations and

formulas that substantiate the developed method.

The material is logically presented, the experimental data are beyond

doubt. However, a number of questions and minor comments arose

for the authors.

1. It remains unclear whether a healthy pancreas or with tumor neoplasms

was used for the ex vivo model?

2. How were the nanoparticles "loaded" into the tissue? Sorption from the

surface, injection into the organ and subsequent preparation of sections?

3. Why was water taken in the study of particles in solution, and not a

biological fluid (for example, serum) or PBS?

4. Table 1. a). Why are there no data for H in values for pancreas?

b). Why in the Au values the value (0.02) for the pancreas is greater than

the GNR-loaded tumor (0.0012)? c). Perhaps there is a typo in the

density value for GNR-loaded tumor (1040012)?

Minor remarks:

Lines 74, 75, 82, 95, 472 Latin terms "in vitro", "in vivo", "in silico",

"ex vivo" should be italicized;

Line 295 "Cabs" must be subscript "abs";

Line 355 “a.l” is replaced by “al.”;

Line 474 item "3.1" to be replaced by "3.2";

Line 504 "12.5 μg/ml." missing "GNRs";

Line 527 "4.5" missing units;

Line 590 "Figure 10D" is missing from the text.

The manuscript may be recommended for publication in the journal

Cancers after making additions.

Author Response

We appreciate the Reviewer for his/her thoughtful evaluation of this manuscript and constructive comments. We have carefully addressed all the comments, aiming at improving the structure and readability of our manuscript. All the revision were highlighted in green colour. 

Manuscript # cancers-2020025

Reviewer’s Comments and Replies

Reviewer 2

  • It remains unclear whether a healthy pancreas or with tumour neoplasms was used for the ex vivo model?

Reply: Thank the Referee for reading the manuscript and for his/her question. Since the purpose was benchmarking the pancreas model used in the simulation (the pancreas without NPs-loaded tumour), we used a healthy pancreas for the ex vivo model.

  • How were the nanoparticles "loaded" into the tissue? Sorption from the surface, injection into the organ and subsequent preparation of sections? 

Reply: Thanks to the Referee for this comment. As it was explained in the section 2.1.4.1., to benchmark the pancreas model simulated with GATE we used a healthy porcine pancreas in the ex vivo study. For benchmarking the simulations in presence of GNRs, we used the water phantom as tissue equivalent phantom to evaluate temperature distribution during 30 s irradiation with laser. We have then compared the experimental results with GATE simulation results, also obtained in in silico water phantoms. This clarification is now reported in the section 2.1.6.1:

“Due to the lack of blood circulation in the ex vivo study, distribution of GNRs cannot be achieved properly in the direct injection of particles to the pancreas. Depending on the type of tissue, clinical applications as well as desired optical properties, there are various phantom matrices such as an aqueous suspensions like water, gelatin or agar-based matrix, and silicone [70]. Here we used water, as it allows controlling the distribution of the NPs to be the most homogeneous as possible.”

  • Why was water taken in the study of particles in solution, and not a biological fluid (for example, serum) or PBS?

Reply: We thank the Referee for this question. Among the tissue equivalent phantoms (like water, agar-based phantom, or silicon, etc,.) for pancreas, we selected water, as all its chemical and physical properties are well known, and we could build a reliable phantom model in the GATE platform.

  • Table 1. a). Why are there no data for H in values for pancreas? b). Why in the Au values the value (0.02) for the pancreas is greater than  the GNR-loaded tumor (0.0012)? c). Perhaps there is a typo in the  density value for GNR-loaded tumor (1040012) “tNAs”?

Reply: We thank the Reviewer for pointing out these oversights. a) In the first line there is information about H and it was now highlighted in green. b) In the row named “ Percent by weight element composition”, bellow the column named “Value for Pancreas) there is “…” means this part doesn’t have any gold composition. 0.0012 is the percent by wight for GNR in the GNR-loaded pancreas tumour. C) when the pancreas is loaded with GNRs the density of tumour is changes and would be more than pancreas without GNRs. 

  • Lines 74, 75, 82, 95, 472 Latin terms "in vitro", "in vivo", "in silico", "ex vivo" should be italicized.

Reply: We thank the Referee for reading our paper carefully. The manuscript has revised according to these comments.

  • Line 295 "Cabs" must be subscript "abs".

Reply – We are thankful for this comment. Cabs is revised to   and refers to cross section for absorption.

  • Line 355 “a.l” is replaced by “al.”.

Reply: Thanks to Referee for this notification. We revised this line.

  • Line 474 item "3.1" to be replaced by "3.2".

Reply: We are thankful for this comment. The revision was done.

  • Line 504 "12.5 μg/ml." missing "GNRs".

Reply: Thanks to the Referee for this notification. The revision was done in this regard.

  • Line 527 "4.5" missing units.

Reply:  Thank you, the unit was added.  

  • Line 590 "Figure 10D" is missing from the text.

Reply – We thank the Referee for this comment. Indeed, it was a typo from our side, as we meant to refer to Figure 9. This part has been corrected in the revised version of the manuscript, as follows:

At 2 mm-distance from the source (Figure 9, yellow lines) and by getting far from the source of 1 cm, in the XY-plane, the temperature increase drops to < 1 °C when the pancreas tumor was loaded with GNRs (Figure 9D, purple line), while it drops to ΔT=6 °C from the at the same place, but in absence of GNRs (Figure 9C, yellow line).

Round 2

Reviewer 1 Report

Accept in present form